# DATA-EFFICIENT IMAGE RECOGNITION WITH CONTRASTIVE PREDICTIVE CODING

## ABSTRACT

Human observers can learn to recognize new categories of objects from a handful of examples, yet doing so with machine perception remains an open challenge. We hypothesize that data-efficient recognition is enabled by representations which make the variability in natural signals more predictable, as suggested by recent perceptual evidence. We therefore revisit and improve Contrastive Predictive Coding, a recently-proposed unsupervised learning framework, and arrive at a representation which enables generalization from small amounts of labeled data. When provided with only 1% of ImageNet labels (i.e. 13 per class), this model retains a strong classification performance, 73% Top-5 accuracy, outperforming supervised networks by 28% (a 65% relative improvement) and state-of-the-art semi-supervised methods by 14%. We also find this representation to serve as a useful substrate for object detection on the PASCAL-VOC 2007 dataset, approaching the performance of representations trained with a fully annotated ImageNet dataset.

## 1 INTRODUCTION

Deep neural networks excel at perceptual tasks when labeled data are abundant, yet their performance degrades substantially when provided with limited supervision (Fig. 1, red). In contrast, humans and animals can quickly learn about new classes of objects from few examples (Landau et al., 1988; Markman, 1989). What accounts for this monumental difference in data-efficiency between biological and machine vision? While highly-structured representations (e.g. as proposed by Lake et al., 2015) may improve data-efficiency, it remains unclear how to program explicit structures that capture the enormous complexity of real visual scenes like those in ImageNet (Russakovsky et al., 2015). An alternative hypothesis has proposed that intelligent systems need not be structured *a priori*, but can instead learn about the structure of the world in an unsupervised manner (Barlow, 1989; Hinton et al., 1999; LeCun et al., 2015). Choosing an appropriate training objective is an open problem, but a promising guiding principle has emerged recently: good representations should make the spatio-temporal variability in natural signals more predictable. Indeed, human perceptual representations have been shown to linearize

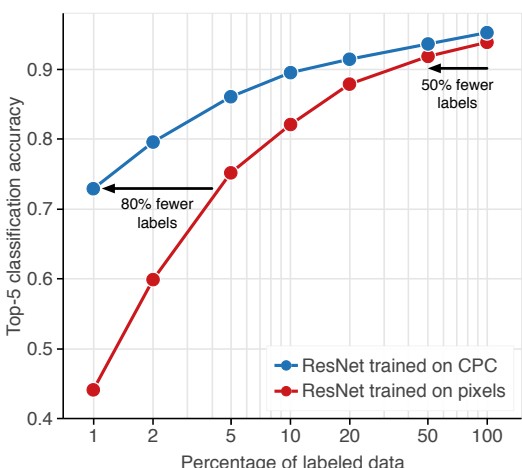

Figure 1: Data-efficient image recognition with Contrastive Predictive Coding. With decreasing amounts of labeled data, supervised networks trained on pixels fail to generalize (red). When trained on unsupervised representations learned with CPC, these networks retain a much higher accuracy in this low-data regime (blue). Equivalently, the accuracy of supervised networks can be matched with significantly fewer labels.

(or 'straighten') the temporal transformations found in natural videos, a property lacking from current supervised image recognition models (Hénaff et al., 2019), and theories of both spatial and temporal predictability have succeeded in describing properties of early visual areas (Rao & Ballard,

1999; Palmer et al., 2015). In this work, we hypothesize that spatially predictable representations may allow artificial systems to benefit from human-like data-efficiency.

Contrastive Predictive Coding (CPC, van den Oord et al., 2018) is an unsupervised objective which learns such predictable representations. CPC is a general technique that only requires in its definition that observations be ordered along e.g. temporal or spatial dimensions, and as such has been applied to a variety of different modalities including speech, natural language and images. This generality, combined with the strong performance of its representations in downstream linear classification tasks, makes CPC a promising candidate for investigating the efficacy of predictable representations for data-efficient image recognition.

Our work makes the following contributions:

- We revisit CPC in terms of its architecture and training methodology, and arrive at a new implementation of CPC with dramatically-improved ability to linearly separate image classes (+17% Top-1 ImageNet classification accuracy).

- We then train deep networks on top of the resulting CPC representations using very few labeled images (e.g. 1% of the ImageNet dataset), and demonstrate test-time classification accuracy far above networks trained on raw pixels (73% Top-5 accuracy, a 28% absolute improvement), outperforming all other unsupervised representation learning methods (+15% Top-5 accuracy over the previous state-of-the-art (Zhai et al., 2019)). Surprisingly, this representation also surpasses supervised methods when given the entire ImageNet dataset (+1% Top-5 accuracy).

- We isolate the contributions of different components of the final model to such downstream tasks. Interestingly, we find that linear classification accuracy is not always predictive of low-data classification accuracy, emphasizing the importance of this metric as a stand-alone benchmark for unsupervised learning.

- Finally, we assess the generality of CPC representations by transferring them to a new task and dataset: object detection on PASCAL-VOC 2007. Consistent with the results from the previous section, we find CPC to give state-of-the-art performance in this setting.

## 2 EXPERIMENTAL SETUP

We first review the CPC architecture and learning objective in section 2.1, before detailing how we use its resulting representations for image recognition tasks in section 2.2.

### 2.1 CONTRASTIVE PREDICTIVE CODING

Contrastive Predictive Coding as formulated in (van den Oord et al., 2018) learns representations by training neural networks to predict the representations of future observations from those of past ones. When applied to images, the original formulation of CPC operates by predicting the representations of patches below a certain position from those above it (Fig. 2, left). These predictions are evaluated using a contrastive loss, in which the network must correctly classify the 'future' representation amongst a set of unrelated 'negative' representations. This avoids trivial solutions such as representing all patches with a constant vector, as would be the case with a mean squared error loss.

In the CPC architecture, each input image is first divided into a set of overlapping patches $x_{i,j}$, each of which is encoded with a neural network $f_\theta$ into a single vector $z_{i,j} = f_\theta(x_{i,j})$. To make predictions, a masked convolutional network $g_\phi$ is then applied to the grid of feature vectors. The masks are such that the receptive field of each resulting *context vector* $c_{i,j}$ only includes feature vectors that lie above it in the image (i.e. $\{z_{u,v}\}_{u \leq i,v}$). The prediction task then consists o predicting 'future' feature vectors $z_{i+k,j}$ from current context vectors $c_{i,j}$, where $k > 0$. The predictions are made linearly: given a context vector $c_{i,j}$, a prediction length $k > 0$, and a prediction matrix $W_k$, the predicted feature vector is $\hat{z}_{i+k,j} = W_k c_{i,j}$.

The quality of this prediction is then evaluated using a contrastive loss. Specifically, the goal is to correctly recognize the target $z_{i+k,j}$ among a set of randomly sampled feature vectors $\{z_l\}$ from the dataset. We compute the probability assigned to the target using a softmax, and evaluate this probability using the usual cross-entropy loss. Summing this loss over locations and prediction

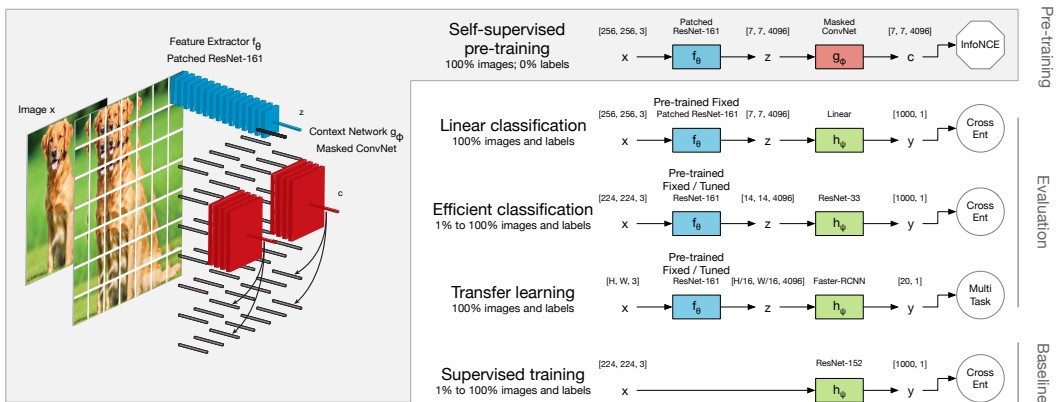

Figure 2: Overview of the framework for semi-supervised learning with Contrastive Predictive Coding. Left: unsupervised pre-training with the spatial prediction task (See Section 2.1). First, an image is divided into a grid of overlapping patches. Each patch is encoded independently from the rest with a feature extractor (blue) which terminates with a mean-pooling operation, yielding a single feature vector for that patch. Doing so for all patches yields a field of such feature vectors (wireframe vectors). Feature vectors above a certain level (in this case, the center of the image) are then aggregated with a context network (red), yielding a row of context vectors which are used to linearly predict features vectors below. Right: using the CPC representation for a classification task. Having trained the encoder network, the context network (red) is discarded and replaced by a classifier network (green) which can be trained in a supervised manner. For some experiments, we also fine-tune the encoder network (blue) for the classification task. When applying the encoder to cropped patches (as opposed to the full image) we refer to it as a *patched* ResNet in the figure.

offsets, we arrive at the CPC objective as defined in (van den Oord et al., 2018):

$$\mathcal{L}_{\text{CPC}} = -\sum_{i,j,k} \log p(\boldsymbol{z}_{i+k,j} | \hat{\boldsymbol{z}}_{i+k,j}, \{\boldsymbol{z}_l\}) = -\sum_{i,j,k} \log \frac{\exp(\hat{\boldsymbol{z}}_{i+k,j}^T \boldsymbol{z}_{i+k,j})}{\exp(\hat{\boldsymbol{z}}_{i+k,j}^T \boldsymbol{z}_{i+k,j}) + \sum_l \exp(\hat{\boldsymbol{z}}_{i+k,j}^T \boldsymbol{z}_l)}$$

The *negative samples* $\{\boldsymbol{z}_l\}$ are taken from other locations in the image and other images in the mini-batch. This loss is called InfoNCE (van den Oord et al., 2018) as it is inspired by Noise-Contrastive Estimation (Gutmann & Hyvärinen, 2010; Mnih & Kavukcuoglu, 2013) and has been shown to maximize the mutual information between $\boldsymbol{c}_{i,j}$ and $\boldsymbol{z}_{i+k,j}$ (van den Oord et al., 2018).

## 2.2 EVALUATION PROTOCOL

Having trained an encoder network $f_\theta$, a context network $g_\phi$, and a set of linear predictors $\{\boldsymbol{W}_k\}$ using the CPC objective, we use the latents $\boldsymbol{z} = f_\theta(\boldsymbol{x})$ as a *representation* of new observations $x$ for downstream tasks, and discard the rest. We then train a model $h_\psi$ to classify these representations given a dataset of labeled images. More formally, given a dataset of $N$ unlabeled images $\mathbb{D}_u = \{x_n\}$, and a (potentially much smaller) dataset of $M$ labeled images $\mathbb{D}_l = \{x_m, y_m\}$:

$$\theta^* = \arg\min_\theta \frac{1}{N} \sum_{n=1}^{N} \mathcal{L}_{\text{CPC}}[f_\theta(x_n)], \quad \psi^* = \arg\min_\psi \frac{1}{M} \sum_{m=1}^{M} \mathcal{L}_{\text{Sup}}[h_\psi \circ f_{\theta^*}(x_m), y_m]$$

In all cases, the dataset of unlabeled images $\mathbb{D}_u$ we pre-train on is the full ImageNet ILSVRC 2012 training set (Russakovsky et al., 2015). We consider three labeled datasets $\mathbb{D}_l$ for evaluation, each with an associated classifier $h_\psi$ and supervised losse $\mathcal{L}_{\text{Sup}}$ (see Fig. 2, right). This protocol is sufficiently generic to allow us to later compare the CPC representation to other methods which have their own means of learning a feature extractor $f_\theta$.

**Linear classification** is the standard benchmark for evaluating the quality of unsupervised image representations. In this regime, the classification network $h_\psi$ is restricted to mean pooling followed by a single linear layer, and the parameters of $f_\theta$ are kept fixed. The labeled dataset $\mathbb{D}_l$ is the entire ImageNet dataset, and the supervised loss $\mathcal{L}_{\text{Sup}}$ is standard cross-entropy. We use the same data-augmentation as in the unsupervised learning phase for training, and none at test time and evaluate with a single crop.

**Efficient classification** directly tests whether the CPC representation enables visual learning from few labels. For this task, the classifier $h_\psi$ is an arbitrary deep neural network (we use an 11-block ResNet architecture with 4096-dimensional feature maps and 1024-dimensional bottleneck layers). The labeled dataset $\mathbb{D}_l$ is a subset of the ImageNet dataset: we investigated using 1%, 2%, 5%, 10%, 20%, 50% and 100% of the ImageNet dataset. The supervised loss $\mathcal{L}_{\text{Sup}}$ is again cross-entropy. In addition to random color-dropping we use the Inception data-augmentation scheme (Szegedy et al., 2014) for training, no augmentation at test-time and evaluate with a single crop.

**Transfer learning** tests the generality of the representation by applying it to a new task and dataset. For this we chose image detection on the PASCAL-2007 dataset, a standard benchmark in computer vision (Everingham et al., 2007). As such $\mathbb{D}_l$ is the entire PASCAL-2007 dataset (comprised of 5011 labeled images); $h_\psi$ and $\mathcal{L}_{\text{Sup}}$ are the Faster-RCNN architecture and loss (Ren et al., 2015). In addition to color-dropping, we use scale-augmentation (Doersch et al., 2015) for training.

For **linear classification**, we keep the feature extractor $f_\theta$ *fixed* to assess the representation in absolute terms. For **efficient classification** and **transfer learning**, we additionally explore *fine-tuning* the feature extractor for the supervised objective. In this regime, we initialize the feature extractor and classifier with the solutions $\theta^*, \psi^*$ found in the previous learning phase, and train them both for the supervised objective. To ensure that the feature extractor does not deviate too much from the solution dictated by the CPC objective, we use a smaller learning rate and early-stopping.

# 3 RELATED WORK

Data-efficient learning has typically been approached by two complementary methods, both of which seek to make use of more plentiful unlabeled data: representation learning and semi-supervised learning. The former formulates an objective to learn a feature extractor $f_\theta$ in an unsupervised manner, whereas the latter directly constrains the classifier $h_\psi$ using the unlabeled data.

**Representation learning** saw early success using generative modeling (Kingma et al., 2014), but likelihood-based models have yet to generalize to more complex stimulus classes. Generative adversarial models have also been harnessed for representation learning (Donahue et al., 2016), and large-scale implementations have recently achieved corresponding gains in linear classification accuracy (Donahue & Simonyan, 2019).

In contrast to generative models which require the reconstruction of observations, self-supervised techniques directly formulate tasks involving the learned representation. For example, simply asking a network to recognize the spatial layout of an image led to representations that transferred to popular vision tasks such as classification and detection (Doersch et al., 2015; Noroozi & Favaro, 2016). Other works showed that prediction of color (Zhang et al., 2016; Larsson et al., 2017) and image orientation (Gidaris et al., 2018), and invariance to data augmentation (Dosovitskiy et al., 2014) can provide useful self-supervised tasks. Beyond single images, works have leveraged video cues such as object tracking (Wang & Gupta, 2015), frame ordering (Misra et al., 2016), and object boundary cues (Li et al., 2016; Pathak et al., 2016). Non-visual information can be equally powerful; information about camera motion (Agrawal et al., 2015; Jayaraman & Grauman, 2015), scene geometry (Zamir et al., 2016), or sound (Arandjelovic & Zisserman, 2017; 2018) can all serve as natural sources of supervision.

While many of these tasks require predicting fixed quantities computed from the data, another class of *contrastive* methods formulate their objectives in the learned representations themselves. CPC is a contrastive representation learning method that maximizes the mutual information between spatially removed latent representations with InfoNCE (van den Oord et al., 2018), a loss function based on Noise-Contrastive Estimation (Gutmann & Hyvärinen, 2010; Mnih & Kavukcuoglu, 2013). Two other methods have recently been proposed using the same loss function, but with different associated prediction tasks. Contrastive Multiview Coding (Tian et al., 2019) maximizes the mutual information between representations of different views of the same observation. Augmented Multiscale Deep InfoMax (AMDIM, Bachman et al., 2019) is most similar to CPC in that it makes predictions across space, but differs in that it also predicts representations across layers in the model. In addition, AMDIM limits the receptive field of its representation, but does this by constraining the number of spatial convolutions in the network architecture rather than using image patches.

A common alternative approach for improving data efficiency is **label-propagation** (Zhu & Ghahramani, 2002), where a classifier is trained on a subset of labeled data, then used to label parts of the unlabeled dataset, after which the process is repeated. This label-propagation can either be discrete (as in pseudo-labeling, Lee, 2013) or continuous (as in entropy minimization, Grandvalet & Bengio, 2005). The predictions of this classifier are often constrained to be smooth with respect to certain deformations, such as data-augmentation (Xie et al., 2019) or adversarial perturbation (Miyato et al., 2018). Representation learning and semi-supervised learning have been shown to be complementary and can be combined to great effect (Zhai et al., 2019), which is why we focus solely on representation learning in this paper.

## 4 RESULTS

When asking whether CPC enables data-efficient learning, we wish to use the best possible representative of this model class. Unfortunately, purely unsupervised metrics tell us little about downstream performance, and implementation details have been shown to matter enormously (Doersch & Zisserman, 2017; Kolesnikov et al., 2019). Since many design choices (e.g. network architecture and data-preprocessing) have been previously evaluated using **linear classification**, we use this benchmark in section 4.1 to align the CPC model with best practices in representation learning and compare to published results. In section 4.2 we select the best performing model from the previous section and assess whether it enables **efficient classification**. We also investigate to what extent the first, more common metric (linear classification accuracy) is predictive of efficient classification. Finally, in section 4.3 we investigate the generality of our results through **transfer learning** to PASCAL-2007.

### 4.1 FROM CPC V1 TO CPC V2

The overarching principle behind our new model design is to increase the scale and efficiency of the encoder architecture while also maximizing the supervisory signal we obtain from each image. At the same time, it is important not to allow the network to solve the problem trivially, i.e., without learning semantics. To this end, we seek to remove low-level cues common across patches by augmenting individual patches independently, using standard stochastic data-processing techniques from supervised and self-supervised learning.

We identify four axes for model capacity and task setup that could impact the model's performance. The first axis increases **model capacity** by increasing depth and width, while the second improves training efficiency capacity by introducing **layer normalization**. The third axis increases task complexity by making **predictions in all four directions**, and the fourth does so by performing more extensive **patch-based augmentation**.

**Model capacity.** Recent work has shown that networks and more effective training improves self-supervised learning (Doersch & Zisserman, 2017; Kolesnikov et al., 2019), but the original CPC model used only the first 3 stacks of a ResNet-101 (He et al., 2016a) architecture (i.e. a ResNet-92). Therefore, we converted the third residual stack of ResNet-101 (originally containing 23 blocks, 1024-dimensional feature maps, and 256-dimensional bottleneck layers), to use 46 blocks, with 4096-dimensional feature maps and 512-dimensional bottleneck layers. We call the resulting network ResNet-161. Consistent with prior results, this new architecture delivers better performance regardless of

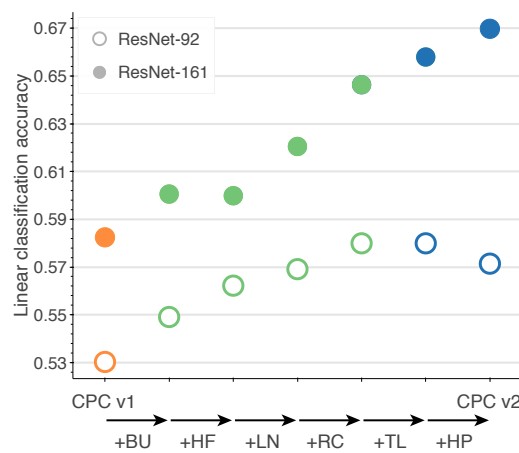

Figure 3: Linear classification performance of new variants of CPC, which incrementally add a series of modifications. BU: bottum up spatial predictions. HF: randomly flipping patches horizontally. LN: layer normalization. RC: random color-dropping. TL: tuned prediction lengths. HP: horizontal spatial predictions. We use color to indicate the number of spatial predictions used (orange, green, blue for 1, 2 and 4 directions).

Table 1: **Linear classifier $h_\psi$ trained with 100% of labels.** Comparison to linear separability of other self-supervised methods. In all cases a feature extractor is optimized in an unsupervised manner, and a linear classifier is trained on top using all labels in the ImageNet dataset.

| Method | Top-1 | Top-5 |
|---|---|---|
| Motion Segmentation (MS) (Pathak et al., 2016) | 27.6 | 48.3 |
| Exemplar (Ex) (Dosovitskiy et al., 2014) | 31.5 | 53.1 |
| Relative Position (RP) (Doersch et al., 2015) | 36.2 | 59.2 |
| Colorization (Col) (Zhang et al., 2016) | 39.6 | 62.5 |
| Combination of MS + Ex + RP + Col (Doersch & Zisserman, 2017) | - | 69.3 |
| CPC v1 (van den Oord et al., 2018) | 48.7 | 73.6 |
| Rotation (Kolesnikov et al., 2019) | 55.4 | - |
| CMC (Tian et al., 2019) | 60.1 | 82.8 |
| Local Aggregation (Zhuang et al., 2019) | 60.2 | - |
| BigBiGAN (Donahue & Simonyan, 2019) | 61.3 | 81.9 |
| AMDIM (Bachman et al., 2019) | 68.1 | - |
| CPC v2 (ours) | 65.9 | 86.6 |

other design choices. Interestingly, a larger architecture delivers larger improvements with more efficient training, more self-supervised losses, and more patch-based augmentations (Fig. 3, **+5%** Top-1 accuracy with original training scheme, **+10%** accuracy with new one).

**Layer normalization.** Large architectures are more difficult to train efficiently. Early works on context prediction with patches used batch normalization (Ioffe & Szegedy, 2015; Doersch et al., 2015) to speed training. However, with CPC we find that batch normalization actually harms downstream performance of large models. We hypothesize that batch normalization allows large models to find a trivial solution to CPC: it introduces a dependency between patches (through the batch statistics) that can be exploited to bypass the constraints on the receptive field. We find that we can reclaim much of batch normalization's training efficiency using layer normalization (Ba et al., 2016), which leads to a small gain for the smaller architecture (**+1%** accuracy over equivalent architectures that use neither normalization) and a larger gain for the larger architecture (**+2.5%** accuracy).

**Prediction lengths and directions.** Larger architectures also run a greater risk of overfitting. We address this by asking more from the network: specifically, whereas van den Oord et al. (2018) predicted each patch using only context from spatially beneath it, we repeatedly predict the patch using context from above, to the right, and to the left, resulting in up to four times as many prediction tasks. Combining top-to-bottom with bottom-to-top helps both model architectures (**+2%** accuracy for both), but using all 4 spatial directions only benefits the larger model (an additional **+1.5%** for the larger model, **-1%** for the smaller), consistent with the idea that model capacity and amount of supervision must go hand-in-hand. We also hypothesized that prediction "length"—i.e. offset between the predicted patch and the aggregated context—might affect performance, as distant patches might lie on distinct objects, encouraging the network to memorize images. Indeed, limiting the range of the prediction length $k$ to $\{2, 3\}$ performed better than $\{2, \ldots, 5\}$ as was used originally (**+1%** for the larger model).

**Patch-based augmentation.** If the network can solve CPC using low-level patterns (e.g. straight lines continuing between patches, chromatic aberration), it need not learn semantically meaningful content. Augmenting the low-level variability across patches can remove such low level cues. The original CPC model spatially jitters individual patches independently. We further this logic by adopting the 'color dropping' method of Doersch et al. (2015), which randomly drops two of the three color channels in each patch, and find it to delivers systematic gains (**+1%** for the small model, **+3%** for the larger one). We also randomly flip patches horizontally, but find it only benefits the smaller model (**+1%**).

**Combined.** Cumulatively, these fairly straightforward implementation changes lead to a substantial improvement to the original CPC model (65.9% Top-1 accuracy, a 17% improvement), making it competitive with recent approaches and outperforming prior methods (see table 1). Interestingly, if we train the same patch-based architecture from scratch in a fully supervised manner, we obtain 66.4% Top-1 accuracy (with batch normalization; 62.5% without), suggesting that CPC is now nearly saturating the architecture's representational power despite not using labels. These results illustrate how architecture and data have an outsized impact on the linear classification performance of self-supervised representations, and are interesting to compare with with previous results. For example, in AMDIM, different settings of data augmentation alone can result in a nearly 10% absolute increase in performance on ImageNet linear classification.

## 4.2 EFFICIENT IMAGE CLASSIFICATION

toWe now turn to our original question of whether CPC can enable data-efficient image recognition. We start by evaluating the performance of purely-supervised networks as the size of the labeled dataset $\mathbb{D}_l$ varies from 1% to 100% of ImageNet, training separate classifiers on each subset. We found that a ResNet-152 to works best across all data-regimes (see Appendix). Despite our efforts to tune the supervised model for low-data classification (including network depth, regularization, and optimization parameters), the accuracy of the best model only reaches 44.1% Top-5 accuracy when trained on 1% of the dataset (compared to 93.9% when trained on the entire dataset, see Fig. 1, red).

**Contrastive Predictive Coding.** We now address our central question of whether CPC enables data-efficient learning. We follow the same paradigm as for the supervised baseline (training and evaluating a separate classifier for each size subset), stacking a neural network classifier on top of the CPC latents $z = f_\theta(x)$ rather than the raw image pixels $x$ (see section 2.2, *efficient classification*, and Appendix). This representation, which we selected for its improved linear classification performance (CPC v2 in Fig. 3), leads to a significant increase in data-efficiency compared to purely supervised networks (Fig. 1, blue curve). This classifier yields 72.9% Top-5 accuracy with only 1% of the labels, a 29% absolute improvement (65% relative) over purely-supervised methods. Surprisingly, when given the entire dataset, this classifier reaches 80.6%/95.2% Top1/Top5 accuracy, surpassing our supervised baseline (ResNet-152: 78.0%/93.9% accuracy) and published results (ResNet-200: 79.9%/95.2%, He et al. (2016b)). We find similar results in all other data-regimes we considered (see Fig. 1).

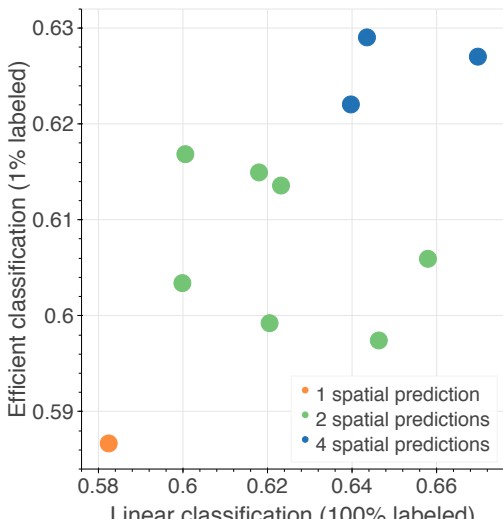

Figure 4: Relationship between linear classification accuracy and low-data classification, for different variants of the CPC model. Left: CPC variants with the same architecture but different training protocols. Orange, green, and blue dots correspond to CPC models making predictions in 1, 2, and 4 spatial directions respectively. Within a color group, different models correspond to other implementation details (e.g. layer norm and patch augmentation, and combinations thereof).

How important are the model specifications described in Section 4.1 for low-data classification? We hypothesized that predictable representations might enable data-efficient classification, and therefore expect that increasing the amount of 'predictability' in the representation should also increase its ability to learn from small amounts of data. Fig. 4 shows evidence for this by ablating model parameters and comparing linear classification performance against low-data classification. Consistent with our hypothesis, increasing the number of spatial directions in the CPC prediction task (which increased linear classification performance) systematically increases low-data classification performance (Fig. 4, left, different color groups). As a control, we asked if all modifications that improve linear classification also improve low-data classification. We did not find evidence in favor of this: improvements in linear classification as a result of changing other model

Table 2: **ResNet classifier $h_\psi$ trained with 1% or 10% of labels.** Comparison to other methods for semi-supervised learning using 1% or 10% of labeled data. *Representation learning* methods use a classifier to discriminate an unsupervised representation, and optimize it solely for the supervised objective on labeled data. *Label-propagation* methods on the other hand further constrain the classifier with smoothness and entropy criteria on unlabeled data, making the additional assumption that all training images fit into a single (unknown) testing category. † denotes methods implemented in this work, *fixed* and *fine-tuned* denote whether the feature extractor is allowed to accommodate the supervised objective.

| Labeled data | 1% | 10% | 100% |
| --- | --- | --- | --- |
| **Method** | **Top-5 accuracy** | | |
| Supervised baseline | 44.1 | 82.1 | 93.9 |
| *Methods using label-propagation:* | | | |
| Pseudolabeling (Zhai et al., 2019) | 51.6 | 82.4 | |
| VAT + Entropy Minimization (Zhai et al., 2019) | 47.0 | 83.4 | |
| Unsup. Data Augmentation (Xie et al., 2019) | - | 88.5 | |
| Rotation + VAT + Ent. Min. (Zhai et al., 2019) | - | **91.2** | 95.0 |
| *Methods using representation learning only:* | | | |
| Instance Discrimination (Wu et al., 2018) | 39.2 | 77.4 | |
| Rotation (Zhai et al., 2019) | 57.5 | 86.4 | |
| †ResNet trained on BigBiGAN (fixed) | 55.2 | 78.8 | 87.0 |
| †ResNet trained on AMDIM (fixed) | 67.4 | 85.8 | 92.2 |
| †ResNet trained on CPC v2 (fixed) | 72.3 | 89.1 | 94.4 |
| †ResNet trained on CPC v2 (fine-tuned) | **72.9** | **89.5** | **95.2** |

parameters (patch-based data-augmentation, layer normalization, and combinations thereof) seem uncorrelated to performance in other tasks (Fig. 4, left, within green group: $R^2 = 0.17, p = 0.36$). Different architectural specifications also produced different changes in both tasks (Fig. 4, right). Whereas increasing the depth of the encoding network greatly improves both metrics, increasing the network width (and therefore the number of features used for linear classification) only improves linear classification accuracy.

**Other unsupervised representations.** How well does the CPC representation compare to other representations that have been learned in an unsupervised manner? If predictable representations are uniquely suited for efficient classification, we would expect other methods within this family to perform similarly, and other model classes less so. Table 2 compares our best model with other works on efficient recognition. We consider three objectives from different model classes: self-supervised learning with rotation prediction (Zhai et al., 2019), large-scale adversarial feature learning (BigBiGAN, Donahue & Simonyan, 2019), and another contrastive prediction objective (AMDIM, Bachman et al., 2019). Zhai et al. (2019) evaluate the low-data classification performance of representations learned with rotation prediction using a similar paradigm and architecture (ResNet-152), hence we report their results directly. Given 1% of ImageNet, their method achieves 57.5% Top-5 accuracy, consistently with the reduced accuracy of a linear classifier (55.4% vs 65.9% for CPC)[1].

Because BigBiGAN and AMDIM achieve stronger linear classification accuracy than rotation prediction (61.3% and 68.1% Top-1 accuracy, respectively), we might expect better performance on efficient classification as well. Since their authors do not report results on efficient classification we evaluated these representations using the same paradigm we used for evaluating CPC, stacking a ResNet classifier on top of the $7 \times 7 \times 8192$ *latents* of the BigBiGAN and the $7 \times 7 \times 2560$ grid of *feature vectors* of AMDIM. We found fine-tuned representations to yield only marginal gains over fixed

---

[1]Although note that linear classification accuracy is reported using a separate model that was designed for linear classification performance (Kolesnikov et al., 2019)

ones (72.9% compared to 72.3% Top-5 accuracy given 1% of labels), hence for simplicity we evaluate BigBiGAN and AMDIM on this task while keeping them fixed. We re-tune the hyper-parameters of the classifier (including optimization, regularization, etc.) for each of these representations separately. Although these methods achieve similar performance in terms of linear classification, we find them to achieve very different results in efficient classification. Given 1% of ImageNet, classifiers trained on top of BigBiGAN achieve 55.2% Top-5 accuracy, similarly to rotation prediction (57.5%), despite its increased linear classification accuracy (+6% relative to rotation prediction). In contrast, AMDIM (which also belongs to the family of contrastive prediction methods) achieves 67.4% on this same task. Again, its increased linear classification accuracy did not entail an increase in data-efficiency. Nevertheless, in line with our initial hypothesis, we find that contrastive prediction methods such as CPC surpass other approaches in our efficient classification experiments, and that linear classification performance is not perfectly correlated with these results.

**Other semi-supervised techniques** A separate class of methods for low-data classification attempts to propagate the knowledge extracted from the subset of labeled examples to unlabeled examples while being invariant to augmentation or other perturbations. These methods generally depend on the quality of the classifier's predictions, and as such tend to fare well when given intermediate amounts of data. Although not sufficient in themselves (Unsupervised Data Augmentation (Xie et al., 2019), Virtual Adversarial Training (Miyato et al., 2018) and entropy minimization (Grandvalet & Bengio, 2005), and pseudo-labeling (Lee, 2013) achieve 85.8%, 83.4%, and 82.4% Top-5 accuracy with 10% of labels, compared to our 89.4%) when combined with representation learning (e.g. rotation prediction Zhai et al., 2019) they can provide considerable gains (91.2% Top-5 accuracy). It is therefore surprising that CPC representations alone can enable accuracy that is comparable to that of these methods, and investigating to what extent they can be combined would be an interesting topic of future work.

### 4.3 TRANSFER LEARNING: IMAGE DETECTION ON PASCAL VOC 2007

We next investigate transfer performance on object detection on the PASCAL-2007 dataset, which reflects the practical scenario where a representation must be trained on a dataset with different statistics than the dataset of interest. This dataset also tests the efficiency of the representation as it only contains 5011 labeled images to train from. In this setting, we replaced the neural network classifier $h_\psi$ used previously with a Faster-RCNN (Ren et al., 2015) image detection architecture, and use the pre-trained feature extractor on ImageNet. As before, we first trained the Faster-RCNN model while keeping the feature extractor fixed, then fine-tuned the entire model end-to-end. Table 3 displays our results compared to other methods. Most competing methods, which optimize a single unsupervised objective on ImageNet before fine-tuning on PASCAL detection, attain around 65% mean average precision. Leveraging larger unlabeled datasets increases their performance up to 67.8% (Caron et al., 2019). Combining multiple forms of self-supervision enables them to reach 70.5% (Doersch & Zisserman, 2017). The proposed method, which learns only from ImageNet data using a single unsupervised objective, reaches 70.6% when equipped with a ResNet-101 feature extractor $f_\theta$ (as for most competing methods (Doersch & Zisserman, 2017) but not all (Caron et al., 2018; 2019). Equipped with the more powerful ResNet-161 feature extractor $f_\theta$, our method reaches 72.7%. Importantly, this result is only 2% short of the performance attained by purely supervised transfer learning, which we obtain by using all ImageNet labels before transferring to PASCAL.

## 5 DISCUSSION

We asked whether CPC could enable data-efficient image recognition, and found that it indeed greatly improves the accuracy of classifiers and object detectors when given small amounts of labeled data. Surprisingly, CPC even improves results given ImageNet-scale labels. Our results show that there is still room for improvement using relatively straightforward changes such as augmentation, optimization, and network architecture. Furthermore, we found that the standard method for evaluating unsupervised representations—linear classification—is only partially predictive of efficient recognition performance, suggesting that further research should focus on efficient recognition as a standalone benchmark. Overall, these results open the door toward research on problems where data is naturally limited, e.g. medical imaging or robotics.

Table 3: **Faster-RCNN $h_\psi$ trained with 100% of PASCAL labels.** Comparison of PASCAL 2007 image detection accuracy to other transfer methods. The supervised baseline learns from the entire labeled ImageNet dataset and fine-tunes for PASCAL detection. The second class of methods learns from the same *unlabeled* images before transferring. All of these methods pre-train on the ImageNet dataset, except for DeeperCluster which learns from the larger, but uncurated, YFCC100M dataset (Thomee et al., 2015). All results are reported in terms of mean average precision (mAP). † denotes methods implemented in this work.

| Method | mAP |
|---|---|
| ***Transfer from labeled data:*** | |
| Supervised - ResNet-152 | 74.7 |
| | |
| ***Transfer from unlabeled data:*** | |
| Exemplar (Ex) (Dosovitskiy et al., 2014) | 60.9 |
| Motion Segmentation (MS) (Pathak et al., 2016) | 61.1 |
| Colorization (Col) (Zhang et al., 2016) | 65.5 |
| Relative Position (RP) (Doersch et al., 2015) | 66.8 |
| Combination of Ex + MS + Col + RP (Doersch & Zisserman, 2017) | 70.5 |
| Instance Discrimination (Wu et al., 2018) | 65.4 |
| Deep Cluster (Caron et al., 2018) | 65.9 |
| Deeper Cluster (Caron et al., 2019) | 67.8 |
| Local Aggregation (Zhuang et al., 2019) | 69.1 |
| †Faster-RCNN trained on CPC v2 (ResNet-101, fine-tuned) | 70.6 |
| †Faster-RCNN trained on CPC v2 (ResNet-161, fine-tuned) | **72.7** |

Furthermore, images are far from the only domain where unsupervised representation learning is important: for example, unsupervised learning is already a critical step in language (Mikolov et al., 2013; Devlin et al., 2018), and shows promise in domains like audio (van den Oord et al., 2018; Arandjelovic & Zisserman, 2018; 2017), video (Jing & Tian, 2018; Misra et al., 2016), and robotic manipulation (Pinto & Gupta, 2016; Pinto et al., 2016; Sermanet et al., 2018). Currently much self-supervised work builds upon tasks tailored for a specific domain (often images), which may not be easily adapted to other domains. Contrastive prediction methods, including the techniques suggested in this paper, are task agnostic and could therefore serve as a unifying framework for integrating these tasks and modalities. This generality is particularly useful given that many real-world environments are inherently multimodal, e.g. robotic environments which can have vision, audio, touch, proprioception, action, and more over long temporal sequences. Given the importance of increasing the amounts of self-supervision (via additional directions of prediction), integrating these modalities and tasks could lead to unsupervised representations which rival the efficiency and effectiveness of biological ones.

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

# A APPENDIX

## A.1 ADDITIONAL RESULTS

Table 4: Data efficient classification results with Top-1 accuracy.

| Labeled data | 1% | 2% | 5% | 10% | 20% | 50% | 100% |
|---|---|---|---|---|---|---|---|
| **Method** | | | | **Top-1 accuracy** | | | |
| Supervised trained on pixels | 23.1 | 34.8 | 50.2 | 60.0 | 67.6 | 74.4 | 78.0 |
| ReseNet trained on CPC v2 (fine-tuned) | 46.3 | 54.7 | 64.1 | 69.7 | 73.0 | 77.3 | 80.6 |

Table 5: Data efficient classification results with Top-5 accuracy (data in Fig. 1).

| Labeled data | 1% | 2% | 5% | 10% | 20% | 50% | 100% |
|---|---|---|---|---|---|---|---|
| **Method** | | | | **Top-5 accuracy** | | | |
| Supervised trained on pixels | 44.1 | 59.9 | 75.2 | 82.1 | 87.9 | 91.8 | 93.8 |
| ReseNet trained on CPC v2 (fine-tuned) | 72.9 | 79.6 | 86.1 | 89.5 | 91.4 | 93.6 | 95.2 |

## A.2 INFONCE IMPLEMENTATION

For completeness, we provide pseudo-code for the main calculations involved in the InfoNCE objective, loosely modeled after Tensorflow operations. We suppose we have just calculated a set of latents $z_{i,j} = f_\theta(x_{i,j})$ for $i, j \in \{1, \ldots, 7\}$, each one being e.g. a 4096-dimensional vector. Assuming we do so for a batch of $B$ images $\{x\}$, the set of latents is a tensor of size $B \times 7 \times 7 \times 4096$.

```
def CPC(latents, target_dim=64, emb_scale=0.1,
        steps_to_ignore=2, steps_to_predict=3):
    # latents: [B, H, W, D]
    loss = 0.0
    context = pixelCNN(latents)
    targets = Conv2D(output_channels=target_dim,
                     kernel_shape=(1, 1))(latents)
    batch_dim, col_dim, row_dim = targets.shape[:-1]
    targets = reshape(target, [-1, target_dim])
    for i in range(steps_to_ignore, steps_to_predict):
        col_dim_i = col_dim - i - 1
        total_elements = batch_dim * col_dim_i * row_dim

        preds_i = Conv2D(output_channels=target_dim,
                         kernel_shape=(1, 1))(context)
        preds_i = preds_i[:, :-(i+1), :, :] * emb_scale
        preds_i = reshape(preds_i, [-1, target_dim])

        logits = matmul(preds_i, targets, transpose_b=True)

        b = range(total_elements) / (col_dim_i * row_dim)
        col = range(total_elements)
        labels = b * col_dim * im_row_dim + (i+1) * row_dim + col

        loss += softmax_cross_entropy_with_logits(logits, labels)
    return loss
```

```python
def pixelCNN(latents):
    # latents: [B, H, W, D]
    cres = latents
    cres_dim = cres.shape[-1]
    for _ in range(5):
      c = Conv2D(output_channels=256,
                 kernel_shape=(1, 1))(cres)
      c = ReLU(c)
      c = Conv2D(output_channels=256,
                 kernel_shape=(1, 3))(c)
      c = Pad(c, [[0, 0], [1, 0], [0, 0], [0, 0]])
      c = Conv2D(output_channels=256,
                 kernel_shape=(2, 1),
                 type='VALID')(c)
      c = ReLU(c)
      c = Conv2D(output_channels=cres_dim,
                 kernel_shape=(1, 1))(c)
      cres = cres + c
    cres = ReLU(cres)
    return cres
```

## A.3 LINEAR CLASSIFICATION

- Model architecture: Having extracted 80x80 patches with a stride of 32x32 from a 240x240 shaped input image, we end up with a grid of 6x6 features (each of which is obtained from our ResNet-161 architecture). This gives us a [6,6,4096] tensor for the image. We then use a Batch-Normalization layer to normalize the features (without scale parameter) followed by a 1x1 convolution mapping each feature in the grid to the 1000 logits for ImageNet classification. We then spatially-mean-pool these logits to end up with the final log probabilities for the linear classification.

- We use the Inception preprocessing (Szegedy et al., 2014) to extract 240x240 crops from the raw image. The image is divided into subcrops as per CPC data-preprocessing used for CPC pre-training.

- Optimization details: We use Adam Optimizer with a learning rate of 5e-4. We train the model on a batch size of 512 images with 32 images per core spread over 16 workers.

## A.4 EFFICIENT CLASSIFICATION: PURELY SUPERVISED

In order to find the best model within this class, we vary the following hyperparameters:

- Model architecture: We investigate using ResNet-50, ResNet-101, and ResNet-152 model architectures, all of them using the 'v2' variant (He et al., 2016b), and find larger architecture to perform better, even when given smaller amounts of data. We insert a DropOut layer before the final linear classification layer (Srivastava et al., 2014).

- Data pre-processing: We use the Inception pre-processing pipeline (Szegedy et al., 2014).

- Optimization details: We vary the learning rate in $\{0.05, 0.1, 0.2\}$, the weight decay logarithmically from $10^{-5}$ to $10^{-2}$, the DropOut linearly from 0 to 1, and the batch size per worker in $\{16, 32\}$.

We chose the best performing model for each training subset $\mathbb{D}_l$ of labeled ImageNet (using a separate validation set), and report its accuracy on the test set (i.e. the publicly available ILSVRC-2012 validation set).

