# OpenReview forum: "Data-Efficient Image Recognition with Contrastive Predictive Coding"
_ICLR.cc/2020/Conference — Reject_

### Official Review · AnonReviewer3 · 2019-10-21
**Official Blind Review #3**

**Rating:** 3

**Review:**

This paper improves Contrastive Predictive Coding method and reaches a good performance in several downstream tasks. However, the novelty and technical contributions are limited.

Strengths:
+ The experimental results seem good. The reimplemented CPC v2 performs much better than the original version. And the performance of down-stream tasks is comparable or better than the state-of-the-art methods.
+ The paper is well written. The paper structure is clear and figures are well illustrated.
+ Figure 3 shows clearly the performance improvements of a series of incremental modifications to the original CPC methods.

Weaknesses:
- The novelty and technical contributions are limited. This paper only proposes some minor improvements based on the original CPC method and use a deeper network to get better performance. The proposed method lacks of important insights for the research community.
- The capacity of network architecture is crucial for self-supervised learning. But in Table 1,2,3, the network architecture of the proposed method is deeper than that in the comparison methods, which is unfair for the comparison methods. Meanwhile, the network architectures of many compared methods are not listed in the tables, which may be misleading. For example, Unsupervised Data Augmentation (Xie et al., 2019) in table 2 and Instance Discrimination (Wu et al., 2018) in table 3 use ResNet50, which is much more shallow than ResNet-161 in this paper.
- In section 2.1, the paper doesn't describe clearly what's the input of masked convolutional network $g_{\phi}$ and how to calculate $c_{i, j}$.



**Experience Assessment:**

I have published in this field for several years.

**Review Assessment: Checking Correctness Of Derivations And Theory:**

I assessed the sensibility of the derivations and theory.

**Review Assessment: Checking Correctness Of Experiments:**

I carefully checked the experiments.

**Review Assessment: Thoroughness In Paper Reading:**

I read the paper thoroughly.

---

> ### Author Response · Authors · 2019-11-13
> **Response to Reviewer #3**
>
> We thank the reviewer for their comments. We respectfully disagree with the assessment that the “novelty and technical contributions are limited”. Although the learning objective we use is the same as in (van den Oord, 2018), we make a number of changes to the training methodology without which the final performance would be uncomparable to the one we arrive at (70.6% Top-1 linear classification accuracy vs 48.7%). We ablate all of these changes and show how important they are for achieving state of the art results. This, combined with the fact that these modifications are very general (and could be straightforwardly applied to audio, video, and text; see footnote), make these technical contributions readily re-usable by the research community. We will open-source our implementation and pre-trained models to make these experimental insights widely accessible.
>
> We agree that it would be interesting to re-evaluate the CPC model with architectures used in other works. These tend to differ widely across papers: (Tian, 2019) use ResNet-101, (Donahue & Simonyan, 2019) use RevNet-50 with 4x width, (Xie, 2019) use ResNet-50 whereas (Zhai, 2019) use ResNet-50 with 4x width. It is therefore difficult to chose a single architecture that will enable comparison to all prior works. Nonetheless, we will systematically list the architectures used by each method and include results from ResNet-50.
>
> The inputs to the masked convolutional network are the feature vectors z_{i,j}. We will make this clear in the text, and provide a reference to the appendix in which this is made explicit.

---

### Official Review · AnonReviewer1 · 2019-10-23
**Official Blind Review #1**

**Rating:** 6

**Review:**

The authors augment contrastive predictive coding (CPC), a recent representation learning technique organized around making local representations maximally useful for predicting other nearby representations, and evaluates their augmented architecture in several image classification problems. Although the modifications to CPC aren't particularly original, the authors show first that these yield a significant improvement in linear classification accuracy. They then use this improved model to obtain impressive performance in classification within semi-supervised and transfer learning settings, giving strong support for the use of such methods within image processing applications.

Pros:
Owing to its generality (CPC assumes only a weak spatial prior in the input data), and cheap computational cost relative to earlier generative approaches, CPC is already a promising unsupervised representation learning technique. The paper gives more evidence of this usefulness for image data, yielding leading performance on several different image classification benchmarks.

The authors also make the observation that linear separability, the standard benchmark for evaluating unsupervised representations, correlates poorly with efficient prediction in the presence of limited labeled data. This observation should be of interest in the broader community, and points to the need for more diverse metrics for unsupervised representations.

Cons:
The improvements given in the paper are quite useful within their stated domain (image data), but aren't directly applicable to other types of input data. Although the authors make a point of emphasizing the relevance of CPC for other problem domains, they don't currently provide any suggestions for how this current work could be generalized to handle these other cases. In this sense, I think it is a bit deceptive to refer to their model as "CPC v2", as the majority of their changes have no bearing on the intrinsic CPC algorithm itself.

I am sure that some of the methods used here could lead to improvements in the use of CPC for other types of data, but the authors currently don't provide any insight on this issue. In line with that, I think their work would be improved by some commentary on this, in particular by any concrete suggestions they have about how similar augmentations to CPC could be carried out in text, audio, and/or video data.

Verdict:
Owing to the reasons given above, I recommend acceptance.

Minor suggestions:
Please use a different color scheme for your figures that is still meaningful if the paper is printed in greyscale.

**Experience Assessment:**

I do not know much about this area.

**Review Assessment: Checking Correctness Of Derivations And Theory:**

N/A

**Review Assessment: Checking Correctness Of Experiments:**

I assessed the sensibility of the experiments.

**Review Assessment: Thoroughness In Paper Reading:**

I read the paper at least twice and used my best judgement in assessing the paper.

---

> ### Author Response · Authors · 2019-11-13
> **Response to Reviewer #1**
>
> We thank the reviewer for their comments. We agree that the modifications we bring to the CPC method are general enough to be applied to a variety of other modalities. For one, the observation that increasing the network depth and ease of optimization can strongly impact performance directly translates to other types of data. Data-augmentation has also become a standard technique in supervised learning, with a considerable amount of domain knowledge being accumulated regarding which techniques are useful for which modalities. Our observation that patch-level augmentation dramatically improves the performance of CPC applied to images could therefore be straightforwardly extended (using analogous augmentation techniques) to audio segments, video cubes, and natural language atoms. Similarly, increasing the number of predictions can easily be applied to other data. As such, since our modifications to the CPC methodology are general enough to be applied to all the modalities for which CPC was originally designed, we think calling it “CPC v2” is valid and warranted, but are curious to hear your suggestions in this matter.
>
> We will make sure to make the figures printer-friendly in the final version.

---

### Official Review · AnonReviewer2 · 2019-11-04
**Official Blind Review #2**

**Rating:** 3

**Review:**

Title: DATA-EFFICIENT IMAGE RECOGNITION
[Summary]
-This paper introduces Contrastive Predictive Coding (CPC) image recognition in the data-efficient regime. Concretely, the authors improve CPC in terms of its architecture and training strategy. The extensive experiments show that CPC enables data-efficient image classification and surpassed other unsupervised approaches.

[Pros]
- Although the CPC was proposed and evaluated in vision task in [1], a new implementation of CPC with dramatically-improved ability is presented in this paper.
- The CPC is utilized to enhance spatially predictable representations which benefits a lot data-efficient image recognition.

[Cons]
- In Sec. 4.1, four axes are identified to upgrade CPC v1 to CPC v2. But they are not well motivated. More discussions about why this four axes are investigated in image recognition.

-The core idea is motivated by a critical hypothesis that good representations should make spatio-temporal variability in natural signals more predictable. However, this hypothesis is not well verified. The concept of amount of ‘predictability’ in page 7 is not clear. It would be great if you provide more evidence that the improvement in low-data classification results from the increased ‘predictability’.

- The comparison in Sec. 4.3 seems unfair. The pretrain model trained with different methods should be the same. For example, the Faster RCNN trained on CPC v2 uses ResNet-101 as backbone but Local Aggregation method uses ResNet-50.

[Summary]
- This work extends CPP to data-efficient image recognition by simply adjusting network architectures and training strategies, which makes it less interesting. Besides, the major hypothesis is not well validated.
- The experimental results are convincing and encouraging. Some minor flaws such as unfair comparison should be fixed.
- I want to see how the four axes in Sec. 4.1 are related to core motivation (more predictable) since they are major adjustments from CPP v1 to CPP v2. If the author provides a profound explanation of the problem, I would consider changing the rating.


**Experience Assessment:**

I have published in this field for several years.

**Review Assessment: Checking Correctness Of Derivations And Theory:**

I assessed the sensibility of the derivations and theory.

**Review Assessment: Checking Correctness Of Experiments:**

I assessed the sensibility of the experiments.

**Review Assessment: Thoroughness In Paper Reading:**

I read the paper at least twice and used my best judgement in assessing the paper.

---

> ### Author Response · Authors · 2019-11-13
> **Response to Reviewer #2**
>
> We thank the reviewer for their comments. Regarding the first point “More discussions about why this four axes are investigated in image recognition,” we agree that a better explanation of the relationship between the new training protocol and our original hypothesis is warranted. Our modifications to the original CPC model can be grouped into 3 categories. Increasing the network scale and ease of optimization both contribute to the representational capacity of the network and its ability to make the complex transformations across space more predictable. The next crucial modification, patch-based augmentation, allows us to control which features of the data will be made more predictable. By making low-level features (such as brightness, color, and contrast) less predictable, we ensure the network capacity is spent on making other features (including the more semantic ones of interest) more predictable. Finally, increasing the number of spatial directions used in the training task amplifies this learning signal. We will update the discussion of these points in section 4.1 to share these intuitions.
>
> Our original hypothesis stated that spatially predictable representations should better enable low-data classification. Through our ablation, we are able to titrate the amount of “predictability” in the representation by changing the number of spatial directions included in the prediction task. For example, one model only attempts to predict patches from top to bottom. The next makes predictions in both vertical directions. The third in all four (horizontal and vertical) spatial directions. These models therefore learn to be “predictable” along more and more axes of the data. In line with our hypothesis, representations which are more “predictable” also enable better low-data classification. However, since these models also improve linear classification accuracy, we asked whether these two metrics were necessarily related to each other. This was not the case (they are uncorrelated across other model specifications, a novel finding in itself), and we therefore take this as evidence that more spatially predictable representations enable efficient classification.
>
> Finally, we agree that it would be interesting to re-evaluate the CPC model with architectures used in other works. Most of the methods we compare to in Table 3 use a ResNet-101, which is why we opted for that architecture. Nevertheless we will include results for ResNet-50 as you suggest in the final version, and report the architecture used by each method.
>
> To conclude, we respectfully disagree with the assessment that this work is “simply adjusting network architectures and training strategies, which makes it less interesting”. Firstly, it is unexpected that the same objective, given a new training protocol, can result in dramatically better performance (from 48.7% to 70.6% linear classification accuracy). Without these results, one might tend to dismiss contrastive learning as impractical or ill-suited to downstream tasks. Furthermore, these modifications are sufficiently general to be applied to a variety of different methods and modalities, and our detailed ablations provide actionable recommendations to the community. We will open-source our implementation and pre-trained models to make these experimental insights widely accessible.

---

### Official Review · AnonReviewer4 · 2019-11-04
**Official Blind Review #4**

**Rating:** 3

**Review:**

The paper proposes to use Contrastive Predictive Coding (CPC), an unsupervised learning approach, to learn representations for further image classification. The authors show that using CPC for representation learning allows to achieve better results than other self-supervised methods. Moreover, CPC is shown to be useful for semi-supervised learning (on par with SOTA method), and transfer learning. All results are very impressive and is in-line with current trends of using a linear classifier on top of a deep feature extractor (e.g., Nalisnick et al., "Hybrid Models with Deep and Invertible Features"). The paper is rather well written and the results are convincing. However, The whole idea of the paper is based on the original paper:
* Oord, Aaron van den, Yazhe Li, and Oriol Vinyals. "Representation learning with contrastive predictive coding." arXiv preprint arXiv:1807.03748 (2018).
Technically speaking, the paper is outstanding, but it lacks novelty in terms of new ideas. I highly appreciate new results and new architectures, but it is not enough for a full conference paper.

Remarks
- In Section 2.1, the problem statement for Contrastive Predictive Coding (CPC) is unclear. For instance, the authors explain CPC by mentioning about masked convolutional layers that is unnecessary at this point. I understand that from engineering perspective it is crucial information, but it does not help to understand CPC. As a result, without knowing the original paper on CPC, Section 2.1 is hard to follow.

- The paper can be treated as an uptaded version of the original CPC paper. I really appreciate all new results and implementation of the idea. The paper is well written and it is technically correct. However, I do not find much novelty compared to the original paper. This would be a perfect workshop contribution, but I am afraid that it is not enough for a full paper.

==== AFTER REBUTTAL ====
I would like to thank the reviewers for their rebuttal. It was not my intention to dismiss your effort in providing new technical results. Please forgive me if you read it in this way. My point is that the paper presents exactly the same idea as the original paper of CPC, but with new, very interesting results. However, I doubt if this is enough for a full conference paper. This point is debatable and I would be happy to further discuss it with other reviewers and the AC. At this point, I keep my original score, but of I am open for a discussion.

**Experience Assessment:**

I have read many papers in this area.

**Review Assessment: Checking Correctness Of Derivations And Theory:**

I assessed the sensibility of the derivations and theory.

**Review Assessment: Checking Correctness Of Experiments:**

I assessed the sensibility of the experiments.

**Review Assessment: Thoroughness In Paper Reading:**

I made a quick assessment of this paper.

---

> ### Author Response · Authors · 2019-11-13
> **Response to Reviewer #4**
>
> We would like to thank the reviewer for their comments on the manuscript. However, we find the decision to dismiss a “technically outstanding” paper simply because it does not introduce a new mathematical formalism to be rather mystifying. Rather than making ever more complex objectives, there is value in reminding the community of the sobering reality that implementation details are hugely important. Dissecting and highlighting the contributions of these details (as we do) will also facilitate the comparison of different self-supervised objectives in future work. To that end, our work makes a number of contributions, both methodological and experimental, which we think will be very impactful to the community.
>
> On the methodological side, we identify a number of axes which enable the performance of CPC: network scale, local data augmentation, amount of self-supervision, etc. These insights are sufficiently general for them to inform other contrastive methods, and other modalities (e.g. audio and video have analogous forms of data-augmentation). Furthermore, it is an important experimental point to notice just how much these “implementation details” matter. Without them, one might dismiss contrastive learning altogether. With them, they appear to be one of the most promising methods for representation learning. We will open-source our implementation and pre-trained models to make these techniques widely accessible.
>
> Moreover, we believe our empirical results to represent a landmark in representation learning: we have shown it to enable substantial gains in data-efficiency for all amounts of available data (as opposed to in only the low-data regime). For the first time, it appears beneficial to train supervised networks on top of learned representations rather than pixel lattices. Going further, our results in transfer learning (which approach that of supervised transfer) raise the possibility of removing the need for large-scale labeled datasets altogether.

---

### Author Response · Authors · 2019-11-13
**New ideas and findings in this work**

We would like to thank the reviewers for their perspective on the manuscript. The main criticism lies with the novelty of our contributions. We disagree with this assessment, for although we do not present any new objective or equation, we present a series of new ideas and findings in this work:

- Representation learning (and CPC in particular) enables unseen gains in the data-efficiency of image classifiers (same performance as purely supervised, with 2-5x less labels) for all amounts of available data (as opposed to only in the low-data regime).
- Representations learned without supervision (with CPC) can rival the performance of supervised representations for transfer learning (to PASCAL).
- The performance of CPC greatly depends on a variety of implementation details, whose contributions we dissect and highlight, providing important insights to the representation learning community. We will open-source our implementation and pre-trained models to make these techniques widely accessible.
- We show that linear classification and low-labeled data classification are not necessarily predictive of each other, motivating the two as independent benchmarks for representation learning.

We identified a number of axes which enable the performance of CPC: network scale, local data augmentation, amount of self-supervision, etc. Following these axes, we have improved our model since the submission, attaining 70.6% Top-1 linear classification accuracy on ImageNet (the original CPC attained 48.7%), setting a new state-of-the-art.

Taken together, our experimental and methodological contributions introduce and defend the idea that representation learning, and CPC in particular, are ready for real-world application.

---

### Decision · Program_Chairs · 2019-12-19

**Decision:**

Reject

**Comment:**

The paper tackles the key question of achieving high prediction performances with few labels. The proposed approach builds upon Contrastive Predictive Coding (van den Oord et al. 2018). The contribution lies in i) refining CPC along several axes including model capacity, directional predictions, patch-based augmentation; ii) showing that the refined representation learned by the called CPC.v2 supports an efficient classification in a few-label regime, and can be transferred to another dataset; iii) showing that the auxiliary losses involved in the CPC are not necessarily predictive of the eventual performance of the network.

This paper generated a hot discussion. Reviewers were not convinced that the paper contributions are sufficiently innovative to deserve being published at ICLR. Authors argued that novelty does not have to lie in equations, and that the new ideas and evidence presented are worth.

The area chair thinks that the paper raises profound questions (e.g., what auxiliary losses are most conducive to learning a good representation; how to divide the computational efforts among the preliminary phase of representation learning and the later phase of classifier learning), but given the number of options and details involved, these results may support several interpretations besides the authors'.

The authors might also want to leave the claim about the generality of the CPC++ principles (e.g., regarding audio) for further work - or to bring additional evidence backing up this claim.

In conclusion, this paper contains brilliant ideas and I hope to see them published with a strengthened analysis of its components.